# The JAK-STAT pathway promotes persistent viral infection by activating apoptosis in insect vectors

Yan Zhang[1,2], Bo-Xue Li[2], Qian-Zhuo Mao[2], Ji-Chong Zhuo[2], Hai-Jian Huang[2], Jia-Bao Lu[2], Chuan-Xi Zhang[2], Jun-Min Li[2], Jian-Ping Chen[1,2]*, Gang Lu[2]*

1 College of Plant Protection, Northwest Agriculture and Forestry University, Yangling, Shaanxi, China,
2 State Key Laboratory for Managing Biotic and Chemical Threats to the Quality and Safety of Agro-products, Key Laboratory of Biotechnology in Plant Protection of MARA and Zhejiang Province, Institute of Plant Virology, Ningbo University, Ningbo, China

* jianpingchen@nbu.edu.cn (J-PC); lugang@nbu.edu.cn (GL)

## Abstract

The Janus kinase-signal transducer and activator of transcription (JAK-STAT) pathway is an evolutionarily conserved signaling pathway that can regulate various biological processes. However, the role of JAK-STAT pathway in the persistent viral infection in insect vectors has rarely been investigated. Here, using a system that comprised two different plant viruses, Rice stripe virus (RSV) and Rice black-streaked dwarf virus (RBSDV), as well as their insect vector small brown planthopper, we elucidated the regulatory mechanism of JAK-STAT pathway in persistent viral infection. Both RSV and RBSDV infection activated the JAK-STAT pathway and promoted the accumulation of suppressor of cytokine signaling 5 (SOCS5), an E3 ubiquitin ligase regulated by the transcription factor STAT5B. Interestingly, the virus-induced SOCS5 directly interacted with the anti-apoptotic B-cell lymphoma-2 (BCL2) to accelerate the BCL2 degradation through the 26S proteasome pathway. As a result, the activation of apoptosis facilitated persistent viral infection in their vector. Furthermore, STAT5B activation promoted virus amplification, whereas STAT5B suppression inhibited apoptosis and reduced virus accumulation. In summary, our results reveal that virus-induced JAK-STAT pathway regulates apoptosis to promote viral infection, and uncover a new regulatory mechanism of the JAK-STAT pathway in the persistent plant virus transmission by arthropod vectors.

## Author summary

The Janus kinase-signal transducer and activator of transcription (JAK-STAT) pathway acts as an innate antiviral immunity in mammals, while its role in the persistent transmission of plant viruses by insect vectors remains largely unknown. In this study, we reported that plant viral infection activated the JAK-STAT pathway and this pathway regulated apoptosis to benefit virus accumulation in the insect vector. This is a new regulatory mechanism of virus-induced apoptosis for persistent viral infection. Moreover, our new

**Data Availability Statement:** The authors confirm that all data underlying the findings are fully available without restriction. All relevant data are

within the paper and its Supporting Information files.

**Funding:** This work was supported by the National Key Research and Development Program of China (No. 2021YFD1401100 to HH), the National Natural Science Foundation of China (No. 32000121 to GL), the Project of the State Key Laboratory for Managing Biotic and Chemical Threats to the Quality and Safety of Agro-products (No. ZS20190102 to JC) and the Ningbo Science and Technology Innovation 2025 Major Project (No. 2019B10004 to JC). The funders had no role in study design, data collection and analysis, decision to publish, or preparation of the manuscript.

**Competing interests:** The authors have declared that no competing interests exist.

findings greatly expand our knowledge on the complex crosstalk between JAK-STAT and other pathways to facilitate persistent virus transmission in their insect vectors.

## Introduction

As a well-studied programmed cell death process, apoptosis exerts a critical function in the pathogenesis of the infectious disease, autoimmunity and cancer [1,2]. The activation of apoptosis can be strictly regulated by an intrinsic mitochondria-mediated or extrinsic death receptor-mediated pathway. Various key events have been concentrated on mitochondria-mediated apoptotic pathway: stimulation of B-cell lymphoma-2 (BCL2) family proteins, permeabilization of mitochondrial outer membrane, the release of mitochondrial factors (e.g., cytochrome *c*), activation of caspase cascade, degradation of cellular macromolecules and trigger of cell death [3–5]. In general, apoptosis is considered to be an efficient defense against viruses in mammals [6–8]. Recently, several arthropod-borne viruses (arboviruses) are found to manipulate the apoptotic pathway to promote viral infection and dissemination in insect vectors [9–13].

The Janus kinase-signal transducer and activator of transcription (JAK-STAT) pathway is a highly conserved signaling pathway in both insects and mammals [14–17]. Activation of the canonical JAK-STAT pathway is initiated by the extracellular binding of polypeptide ligands to specific transmembrane receptors. As a result, the receptor undergoes a conformational transformation and self-phosphorylation of the receptor-associated Janus kinases (JAKs). The activated JAKs in turn phosphorylate the receptor, contributing to the formation of docking sites for cytoplasmic signal transducer and activator of transcriptions (STATs). Then, the phosphorylated STATs are recruited under the action of JAK, which ultimately translocate into the nucleus where they activate the transcription of target genes [18,19]. The JAK-STAT pathway has been reported to participate in a number of vital biological processes, including growth regulation [20], hematopoiesis [21], oncogenesis [22,23], embryonic development [24] and innate antiviral immunity [17,25]. The suppressor of cytokine signaling (SOCS) family, induced by JAK-STAT signaling cascade, plays a key regulatory role in antiviral immunity through suppressing of signal transducer activities or proteasome degradation of signaling proteins [26]. Moreover, increasing studies have indicated that SOCS family may mediate the crosstalk between JAK-STAT and other pathways [27–30].

Even though persistent arboviruses have seriously threatened the host health globally, they cause low or even no pathological symptoms in insect vectors during virus acquisition and transmission [31–33]. Arboviruses can activate multiple immune responses in insects, including cellular and humoral immune, RNA interference and PCD response, aiming to overcome tissue barriers and promote persistent infection [32,33]. Among them, virus-induced innate signaling pathways play vital roles in maintaining a balance between viral accumulation and vector fitness. For example, RNA interference pathway maintains arboviral infection and vector competence for transmission [34–36]. Activation of mitogen-activated protein kinase (MAPK) signaling pathway affects viral infection and regulates PCD immune response [37–39]. Besides, c-Jun N-terminal kinase (JNK) pathway has been shown with a broad antiviral function against viral infection in *Aedes aegypti* [40]. Conversely, a plant virus manipulates JNK pathway to promote viral replication in its insect vector [41]. Recently, JAK-STAT pathway has been reported to maintain a balance between vector fitness and virus transmission [42]. Whether virus-induced innate immune system is involved in apoptotic response during persistent viral infection remains largely unclear.

The small brown planthopper (SBPH, *Laodelphax striatellus*) is a notorious insect vector, which transmits multiple plant RNA viruses, including Rice stripe virus (RSV, *Phenuiviridae*) and Rice black-streaked dwarf virus (RBSDV, *Spinareoviridae*), in a persistent-propagative manner [43,44]. Previous results have demonstrated that JNK pathway and AMP-activated protein kinase (AMPK) pathway can be activated by RSV and RBSDV, and participate in the persistent viral infection in SBPH [41,45]. In this study, we found that the infection by RSV and RBSDV activates JAK-STAT pathway and this pathway regulates apoptosis to promote persistent viral infection in SBPH vectors.

## Results

### Viral infection induces apoptosis in insect vectors

To explore whether apoptosis is induced by RSV infection, we first examined the morphology of midgut epithelial cells from viruliferous or nonviruliferous SBPH vectors. Under transmission electron microscopy (TEM), abundant apoptotic cells were observed in RSV-infected SBPHs. Compared with intact nuclei and finely dispersed chromatin in nonviruliferous cells, these virus-infected epithelial cells displayed apoptotic characteristics, including condensed nuclei and marginalized chromatin. Meanwhile, the vacuolated mitochondria and irregular cristae were detected in virus-infected but not in RSV-free midgut epithelial cells (Fig 1A). To count the number of apoptotic cells, approximately 200 epithelial cells in nonviruliferous or viruliferous samples were observed using TEM. Statistical analysis suggested that the percentage of apoptotic cells in the viruliferous was significantly higher than in the nonviruliferous SBPHs (Fig 1B). Furthermore, terminal uridine nick-end labeling (TUNEL) assay was performed to detect DNA fragmentation in SBPH midguts and salivary glands. Compared with the RSV-free cells, large numbers of crescent-shaped nuclei (blue) were observed in RSV-infected cells (Fig 1C and 1E). Besides, more positive apoptotic signals (green) were detected in RSV-infected midguts and salivary glands of viruliferous than those in RSV-free SBPHs (Fig 1C–1F).

To confirm whether mitochondria pathways were involved in RSV-induced apoptosis, we first examined the expression of two key factors of mitochondrial-dependent apoptosis, apoptotic protease activating factor-1 (APAF-1) and apoptosis-inducing factor (AIF) in nonviruliferous and viruliferous SBPHs by western blotting. The protein levels of APAF-1 and AIF increased in viruliferous compared with nonviruliferous SBPHs (Fig 1G, lanes 1–3 vs. 4–6). Afterwards, the release of pro-apoptotic cytochrome *c* (Cyt C) and the degradation of anti-apoptotic BCL2 could be detected. The accumulation of Cyt C was much higher and the BCL2 significantly decreased in viruliferous than in nonviruliferous insects (Fig 1G, lanes 1–3 vs. 4–6). The activation of mitochondria-dependent apoptosis was further confirmed during RSV infection. Consistent with previous results, after the RSV crude extracts from viruliferous SBPHs were injected into the hemocoel of nonviruliferous SBPHs, the protein levels of APAF-1 and AIF also increased compared with the mock-infected group. Meanwhile, an obvious accumulation of Cyt C and remarkable consumption of BCL2 following RSV infection were observed (Fig 1H, lanes 1–3 vs. 4–6). These results strongly suggested that RSV infection induces apoptotic response in insect vectors.

### Inhibition of apoptosis reduces viral accumulation in insect vectors

After determining that RSV infection is capable of inducing apoptosis, we then investigated the effects of the apoptotic pathway on viral infection in insect vectors. Four key factors, containing Caspase1a (Casp1a), Caspase1c (Casp1c), Caspase8 (Casp8) and CaspaseNc (CaspNc), have been reported to be involved in the mitochondria-mediated apoptotic pathway [13,46].

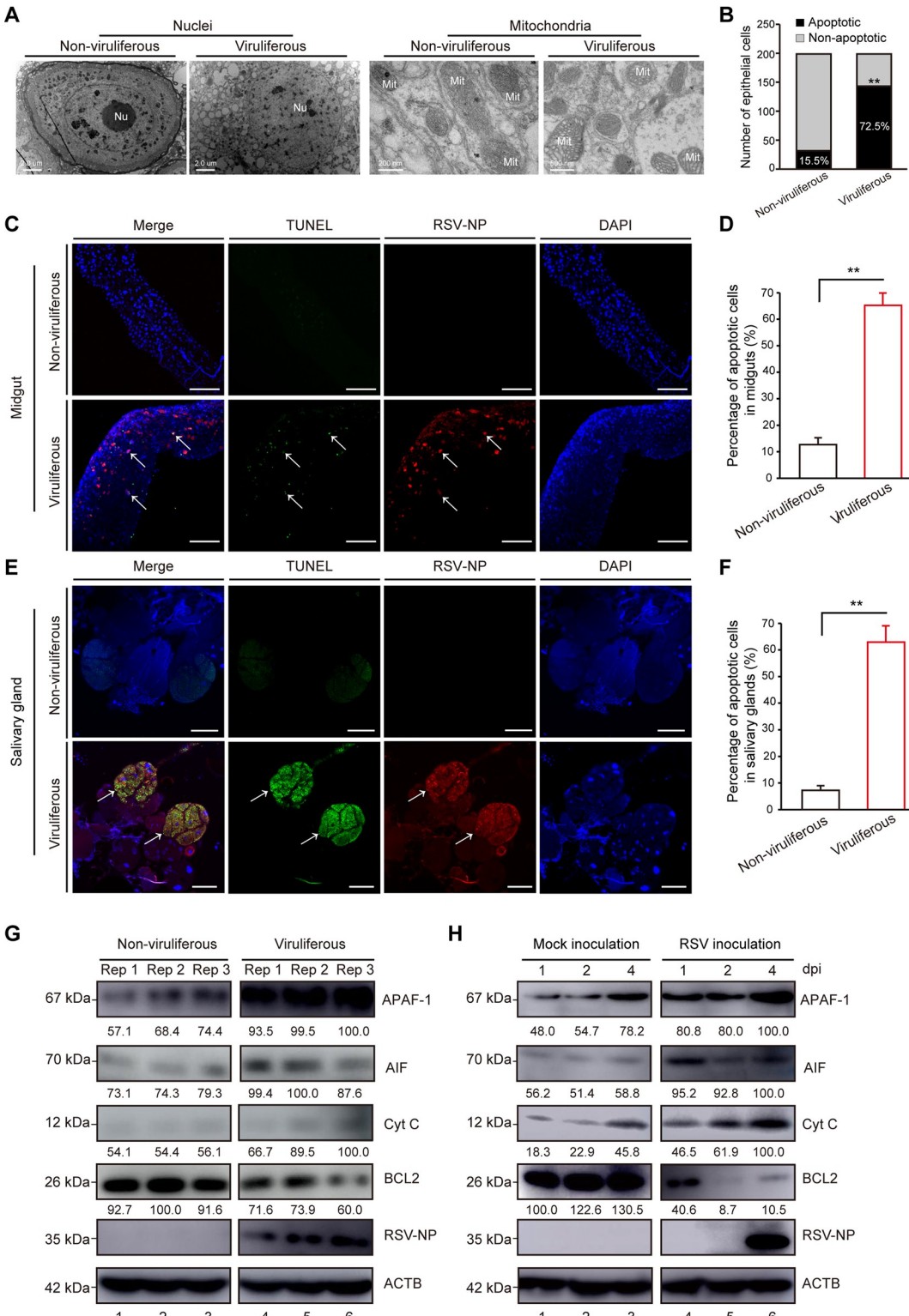

**Fig 1. Viral infection induces apoptosis in insect vectors.** (A) The nuclei and mitochondria of midgut epithelial cells from nonviruliferous and viruliferous SBPHs were detected using TEM. Nu, nucleus. Mit, mitochondrion. (B) Biological statistics of apoptotic cells from nonviruliferous and viruliferous SBPHs. **, p < 0.01 by the student t-test. (C–F) TUNEL assay showing midgut cells (C) and salivary gland cells (E) of nonviruliferous and viruliferous SBPHs. Apoptosis cells were labeled with TUNEL (green), RSV-NP (red) and DAPI straining (blue). Bar, 100 μm. Biological statistics of apoptotic cells from

midguts (D) and salivary glands (F). The experiment was replicated three times. **, p < 0.01 by the student t-test. (G) Analysis of apoptotic response in nonviruliferous and viruliferous SBPHs using Western blotting assay with the indicated antibodies. Three independent repetition samples (Rep 1, Rep 2 and Rep 3) from nonviruliferous or viruliferous SBPHs were detected for each antibody. (H) Analysis of apoptotic response in SBPHs during the time course of RSV infection. Injection of nonviruliferous SBPHs crude extracts as a mock inoculation. ACTB was used as a protein loading control.

These genes were first identified in the transcriptome data from SBPH and phylogenetic analyses were performed based on their conserved domains. Results showed that these apoptotic factors were conserved with other insect species (S1 Fig). Using RNA-mediated interference (RNAi), the expression of four apoptotic factors was significantly reduced (Fig 2A, 2C, 2E and 2G). After the protein level of Cyt C decreased, the accumulation of viral nucleocapsid protein (NP) at RNA level or protein level was reduced significantly at 4 dpi compared with the control group that was injected with double-stranded RNA for green fluorescent protein (dsGFP) (Fig 2B, 2D, 2F, 2H and 2I). Next, SBPHs were fed with an apoptosis inhibitor (Z-DEME-FMK) or an apoptosis activator (PAC-1) using artificial double parafilm for 48 h after being injected with RSV crude extracts. Compared with dimethyl sulfoxide (DMSO)-treated control, the accumulation of viral NP significantly decreased after treatment with Z-DEME-FMK, while significantly increased after treatment with PAC-1 (Fig 2J). According to western blotting, feeding SBPH with PAC-1 was confirmed to induce the expression of apoptosis-related factors (APAF-1, AIF, Cyt C) (S2 Fig). As expected, immunofluorescence labeling using the RSV NP specific antibody suggested that fewer RSV virions were observed in dsCasp1a-treated SBPH salivary glands and ovaries compared with dsGFP-treated samples at 4 dpi (Fig 2K–2M). Collectively, these results demonstrate that inhibition of apoptosis leads to a reduction of viral accumulation in SBPH vectors.

## Viral infection promotes the degradation of BCL2 via the SOCS5-mediated proteasome pathway

Anti-apoptotic BCL2 has been reported to play a key regulatory role in mitochondrial apoptosis signaling and the degradation of BCL2 can trigger the apoptotic cascade [47,48]. Considering that viral infection promotes the degradation of BCL2 (Fig 1G and 1H), we examined which protein is involved in this process. Yeast two-hybrid (Y2H) system was therefore performed to screen the target proteins in insect cDNA library using BCL2 as the bait. A putative suppressor of cytokine signaling protein 5 (SOCS5) was selected due to its possible function in apoptosis [26,49,50]. The phylogenetic tree and sequence analysis showed that this SOCS5, encoding a conserved SH2 and a SOCS box domain, was clustered with that of other insects (Figs 3A and S3). Using one-to-one Y2H assay and pull-down assay, the interaction between BCL2 and SOCS5 was further confirmed (Fig 3B and 3C). In order to determine which domain in SOCS5 interact with BCL2, SOCS5 was divided into four fragments: SOCS5(N) (N-terminus, 1–250aa), SOCS5(C) (C-terminus, 251–497aa), SOCS5(SH2) (SH2 domain, 335–435aa) and SOCS5(SOCS) (SOCS box domain, 436–497aa) (Fig 3A). Both one-to-one Y2H assay and MBP pull-down assay indicated that SOCS5 interact with BCL2 via the SH2 domain (Fig 3B and 3C).

To better understand the role of SOCS5 on the expression of BCL2, dsSOCS5 was injected into the hemolymph of viruliferous third-instar SBPHs. The accumulation of BCL2 significantly increased, and RSV NP was reduced notably compared with the dsGFP-treated group (Fig 3D, lanes 3 vs. 4). In addition, no obvious effect on protein level accumulation of BCL2 was detected after treating nonviruliferous SBPH with dsSOCS5 compared with the control

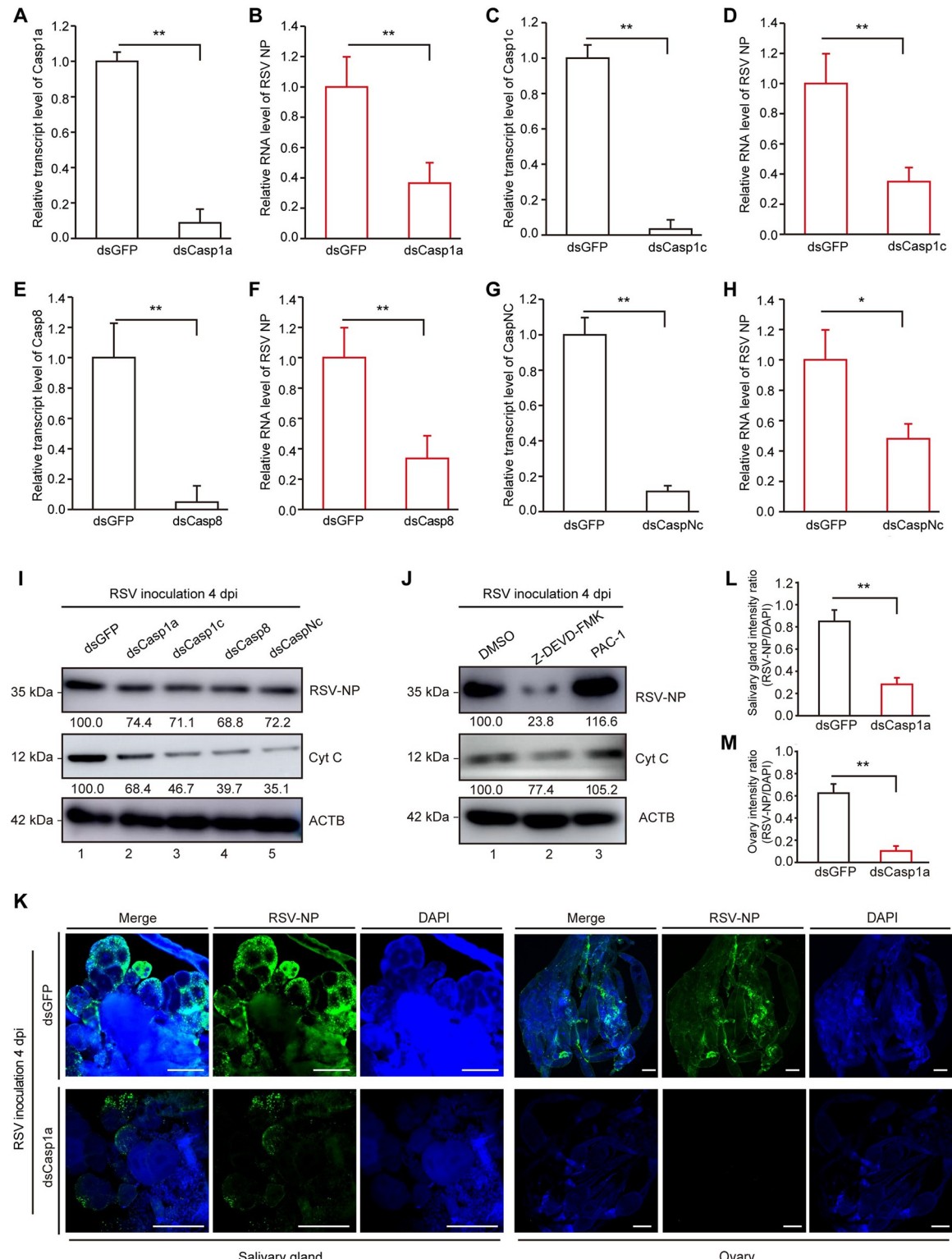

**Fig 2. Effects of apoptosis on virus accumulation in insect vectors.** After silencing of four apoptotic factors, [Casp1a (A), Casp1c (C), Casp8 (E) and CaspNc (G)], the accumulation of RSV NP at RNA level (B, D, F, H) and protein level (I) were detected at 4 dpi. (J) Western blotting analysis of RSV NP and Cyt C in SBPHs that fed with apoptosis inhibitor Z-DEME-FMK or apoptosis activator PAC-1. SBPHs that fed with DMSO were acted as the control group. (K) Immunofluorescence labeling of RSV NP (green) in salivary glands and ovaries from ds*Casp1a*- or ds*GFP*-treated SBPHs at 4 dpi. Nuclei were stained with DAPI (blue). Bar, 100 μm. (L–M) Fluorescence

density of RSV NP signal in salivary glands (L) and ovaries (M). Relative intensity was quantified by ImageJ software. The experiment was replicated three times. **, p < 0.01 by the student t-test.

(Fig 3D, lanes 1 vs. 2), suggesting that SOCS5 imposes a negative effect on BCL2 accumulation during viral infection.

The 26S proteasome system refers to one of the major pathways for protein degradation. To confirm its function in the SOCS5-induced reduction of BCL2 protein, proteasome inhibitor MG132 was treated with nonviruliferous and viruliferous SBPHs using artificial double parafilm feeding for 48 h. The protein levels of BCL2 or SOCS5 were similar between MG132-treated and DMSO-treated groups in nonviruliferous SBPHs (Fig 3E, lanes 1 vs. 2). Nevertheless, the BCL2 protein levels were recovered, and the accumulation of RSV NP was substantially reduced in MG132-treated viruliferous SBPHs compared with the control group (Fig 3E, lanes 3 vs. 4), suggesting that the proteasome pathway is involved in SOCS5-induced degradation of BCL2.

To further investigate whether the SOCS5-induced degradation of BCL2 is physically associated with mitochondria, we performed mitochondria isolation assay. Western blotting analyses revealed that the accumulation of BCL2 or SOCS5 was mainly found in the mitochondria fraction, but not in cytosol fraction (Fig 3F). Meanwhile, the accumulation of BCL2 was inhibited and SOCS5 protein levels increased in mitochondria of viruliferous compared with the nonviruliferous groups (Fig 3F, lanes 5–6 vs. 7–8), indicating that SOCS5-induced inhibition of BCL2 accumulation is associated with mitochondria.

## Viral infection activates the JAK-STAT pathway in insect vectors

The gene expression of SOCS family has been reported to be regulated by JAK-STAT pathway in mammals [51,52]. To investigate whether STAT5B directly regulate the expression of *SOCS5* gene in SBPHs, we first identified canonical STAT5B-binding site (TTCN2–4GAA) in the promoter region of *SOCS5* (3 kb upstream region) based on the genome database of SBPH [53,54]. Three candidate STAT5B-binding sites (SBSs) were selected, namely, SBS1 (-120 to -113, TTCTTGAA), SBS2 (-1871 to -1862, TTCAATGGAA) and SBS3 (-2946 to -2938, TTCTCTGAA) in the upstream promoter sequence of *SOCS5* gene (Fig 4A). We then performed yeast one-hybrid (Y1H) and electrophoretic mobility shift assay (EMSA) assays to confirm the direct binding of STAT5B to the three SBSs. In the Y1H assays, yeast cells co-transformed with STAT5B and SBS1, SBS2 or SBS3, respectively, grew normally in the selective medium, whereas yeast cells containing the empty vector (EV) and three SBS sequences did not grow (Fig 4B). In the EMSA assays, STAT5B was also found to bind to three SBS probes (Fig 4C). Later, we tested the effect of other factors in JAK-STAT pathway on the expression of *SOCS5* gene. The nonviruliferous SBPHs were treated with ds*JAK* or ds*GFP* for 48 h. Western blotting analyses revealed that the protein expression of SOCS5 was reduced in ds*JAK*-treated SBPHs compared with the ds*GFP*-treated control (S4A Fig). These results thus demonstrated that the *SOCS5* is likely to be regulated by JAK-STAT pathway.

To explore whether the JAK-STAT pathway is activated upon viral infection, we first identified six core genes (*JAK*, tyrosine kinase 2 [*TYK2*], STAT5B, signal-transducing adapter molecule [*STAM*], tyrosine phosphatase non-receptor type 2 [*PTPN2*], protein inhibitor of activated STAT [*PIAS*]) in this pathway based on insect transcriptome data. Phylogenetic analyses showed that these genes were conserved with other insect species (S5 Fig). Compared with nonviruliferous group, these core genes did not exhibit any obvious variation at the transcriptional level in viruliferous SBPHs (S4B Fig). However, the phosphorylation of transcription factor STAT5B (P-STAT5B) was obviously detected in viruliferous but not in

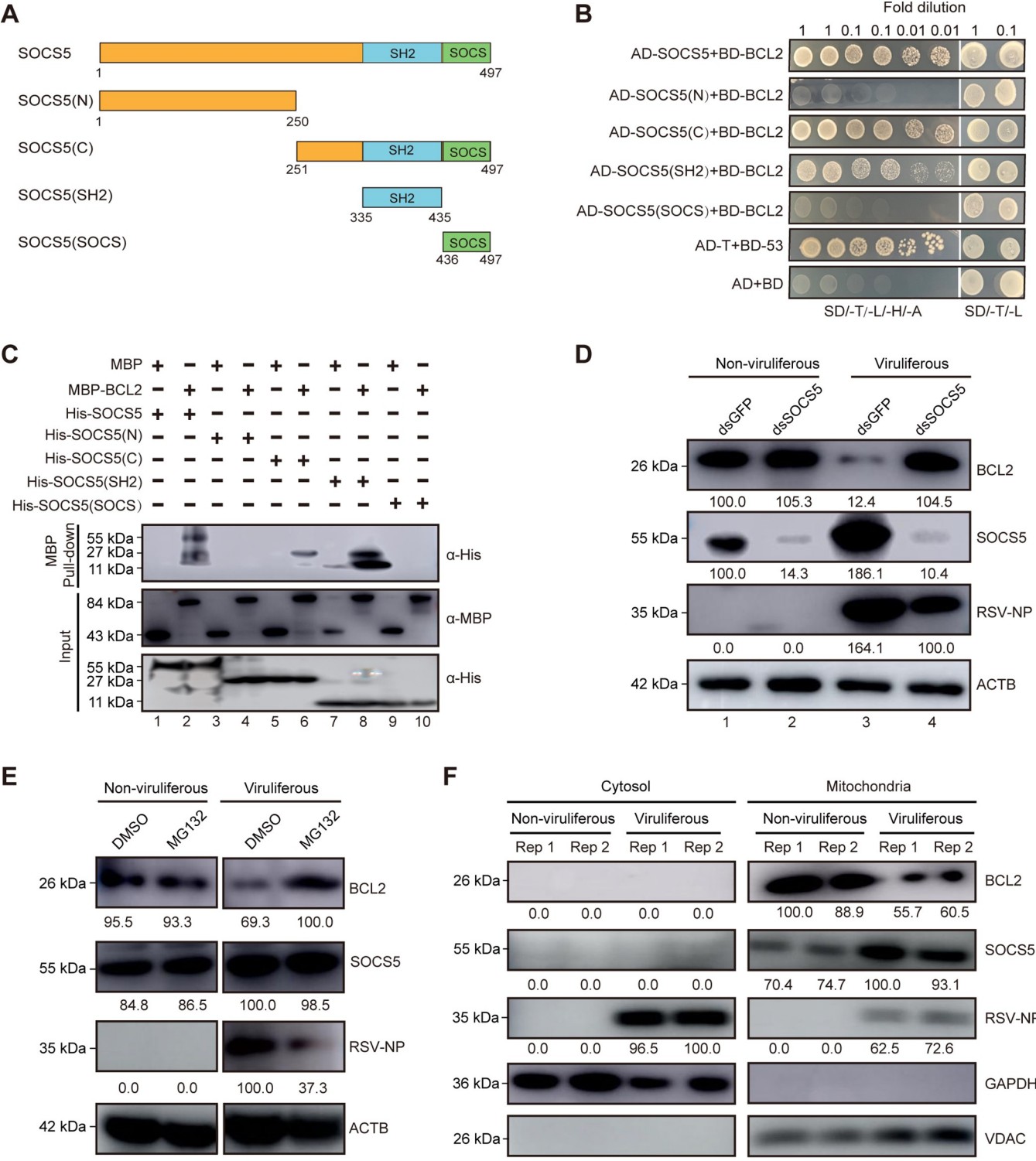

**Fig 3. SOCS5 promotes the degradation of BCL2 during viral infection.** (A) Structure feature of full-length and four fragments of SOCS5. (B) SH2 domain of SOCS5 specifically interacted with BCL2 as detected by yeast two-hybrid assay. The co-transformed AH109 yeast cells with 10-fold serial dilutions were plated on synthetic dextrose (SD) lacking tryptophan, leucine, histidine, and adenine (SD/-T/-L/-H/-A) or SD/-T/-L medium to analyze positive interactions. Yeast cells co-transformed with AD-T+BD-53 served as a positive control. Yeast co-transformed with AD+BD acted as a negative control. (C) MBP pull-down assay of the interaction between SOCS5 and BCL2. MBP, MBP-BCL2, full-length and fragments of His-SOCS5 were expressed and purified from the *E. coli*. The purified MBP or MBP-BCL2 was individually mixed with full-length and four fragments of His-SOCS5 and pulled down with MBP beads. MBP-BCL2

pulled down His-SOCS5 (Lane 2), His-SOCS5(C) (Lane 6) and His-SOCS5(SH2) (Lane 8) using His antibody. The mixture samples before MBP pull-down (Input) were detected by immunoblotting using MBP or His antibody. (D) Immunoblotting analysis of the degradation of BCL2 after injection of ds*SOCS5* or ds*GFP* into nonviruliferous and viruliferous with the indicated antibodies. (E) Immunoblotting analysis of the degradation of BCL2 after feeding nonviruliferous and viruliferous SBPHs with DMSO or MG132 for 4 days. (F) Immunoblotting analysis of the protein levels of SOCS5 and BCL2 in mitochondria and cytosol fractions. The glyceraldehyde-3-phosphate dehydrogenase (GAPDH) was used as a cytosol marker and the voltage-dependent anion channel (VDAC) was used as a mitochondria marker. ACTB was used as a protein loading control.

nonviruliferous SBPHs. In addition, the protein expression of SOCS5 increased significantly in viruliferous than that in nonviruliferous SBPHs (Fig 4D). Next, after RSV crude extracts were injected into the hemolymph of nonviruliferous SBPHs, the protein levels of SOCS5 and phosphorylated STAT5B gradually increased during the course of viral infection (Fig 4E). Similarly, the mRNA levels of SOCS5 and STAT5B were both up-regulated at 1 dpi, but restored at 2 or 4 dpi, compared with mock inoculation controls (Fig 4F–4I). These results suggested that RSV infection activates the insect JAK-STAT pathway and leads to the accumulation of SOCS5 protein.

## JAK-STAT pathway regulates apoptosis to promote viral infection

To determine the role of JAK-STAT pathway in viral infection, we first knocked down the expression of *STAT5B* and *SOCS5* using RNAi in viruliferous SBPHs. After decreasing in the relative expression of *STAT5B* and *SOCS5*, a significant decrease in viral accumulation in ds*STAT5B*- or ds*SOCS5*-treated SBPHs was observed compared with the control group (Fig 5A and 5B). In addition, nonviruliferous SBPHs were treated with ds*STAT5B*, ds*SOCS5* and ds*GFP*, respectively, for 2 days followed by injection of RSV crude extracts. The results showed that silencing of *STAT5B* or *SOCS5* remarkably reduced viral genomic RNAs (RNA3 and RNA4) compared with the control at 2 and 4 dpi (Fig 5C–5E). Similar to the above results, immunolabeling using a NP specific antibody showed that ds*STAT5B*- or ds*SOCS5*-treated SBPHs resulted in fewer RSV virions in salivary glands and ovaries compared with ds*GFP* treatment at 4 dpi (Fig 5F–5H). Furthermore, ds*STAT5B*- or ds*SOCS5*-treated SBPHs had drastically reduced virus acquisition compared with the ds*GFP*-treated group after injection of RSV crude extracts (Fig 5H and 5I). In another group, ds*STAT5B*- or ds*SOCS5*-treated SBPHs were allowed to feed on RSV-infected rice plants for 48 h and transferred to healthy rice plants for another 8 days. RSV acquisition rates in ds*STAT5B* and ds*SOCS5*-treated SBPHs were significantly lower than that in ds*GFP*-treated group (S6A Fig). Meanwhile, ds*STAT5B*-treated insects exhibited no significant effect on their survival during RSV infection, while the mortality of ds*SOCS5*-treated insects was reduced remarkably compared with the ds*GFP* control at 6 and 8 days post infection (S6B Fig).

We further investigated whether the activation of JAK-STAT pathway affects apoptosis. After silencing of *STAT5B* or *SOCS5* using RNAi, the protein levels of key factors of mitochondrial-dependent apoptosis, such as Cyt C, APAF-1 and AIF, were substantially reduced and the accumulation of BCL2 significantly increased at 4 dpi. Additionally, the accumulation levels of viral NP were reduced remarkably in ds*STAT5B* and ds*SOCS5* treatment compared with the control group (Fig 5K). Our data suggested that inhibition of JAK-STAT pathway activation contributes to the reduction of apoptosis and suppression of RSV infection in SBPHs. Collectively, JAK-STAT pathway can regulate apoptosis to promote viral infection in SBPHs.

## JAK-STAT pathway promotes other plant viral infection in insect vectors

To further investigate the effect of JAK-STAT pathway on other arbovirus, another SBPH-transmitted plant reovirus, Rice black streaked dwarf virus (RBSDV), was inoculated into

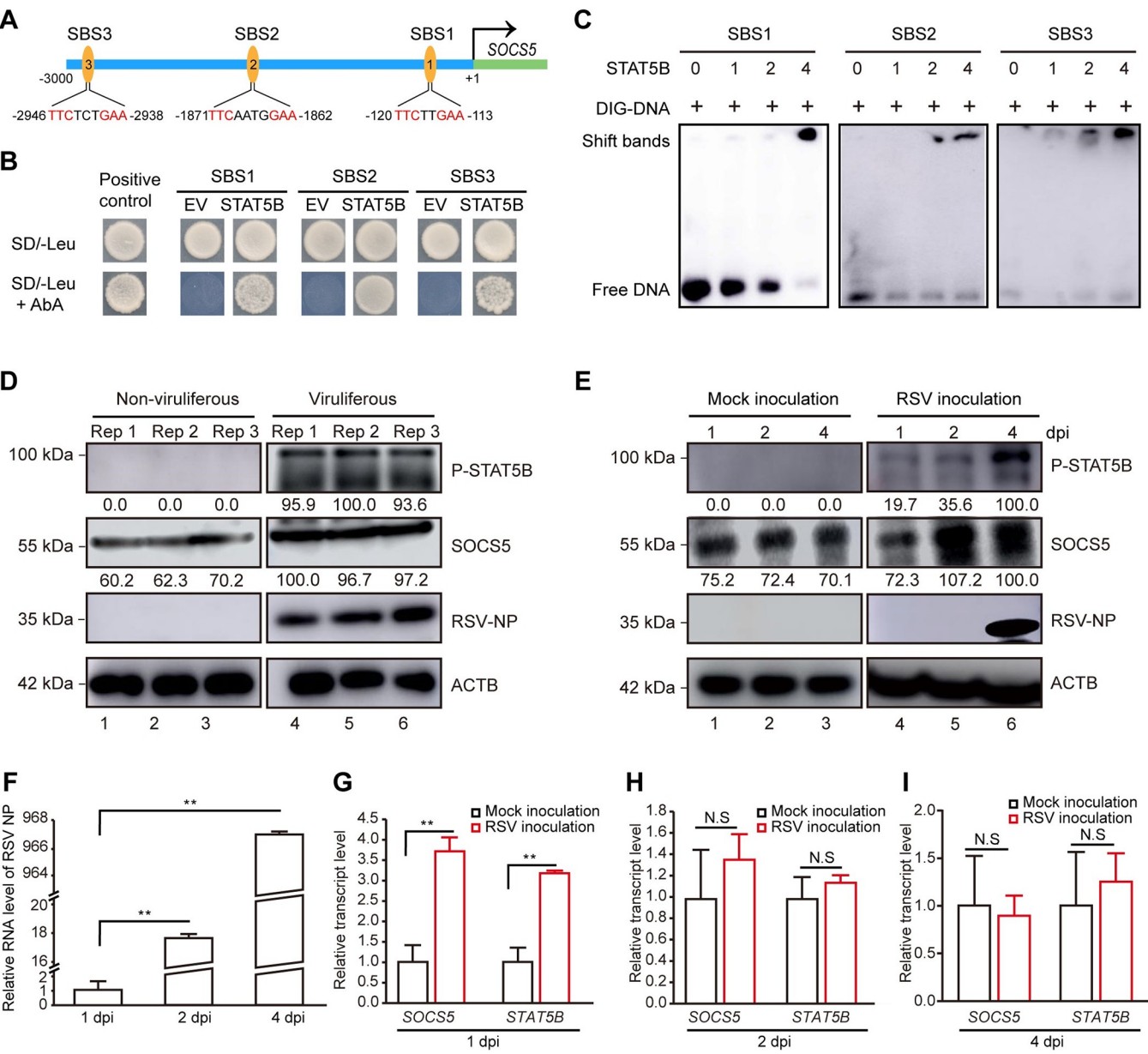

**Fig 4. Viral infection activates the JAK-STAT pathway in insect vectors.** (A) Schematic diagram showing three putative STAT5B-binding sites (SBS1, SBS2 and SBS3) in the promoter region of *SOCS5* gene. (B) Y1H assay verifying the binding of STAT5B to the three binding sites in the *SOCS5* promoter. The co-transformed yeast cells were grown on synthetic dextrose (SD) with 25ng/mL AbA lacking leucine (SD/-L+AbA) medium for 4 days at 30˚C. Yeast cells co-transformed with AD-p53+pABAi-p53 acted as a positive control. EV, empty vector. (C) EMSA assay confirming the direct binding of STAT5B to the three binding sites in the *SOCS5* promoter. Increasing amounts of STAT5B were incubated with 60 fmol of three DIG-labeled promoters, respectively. (D) Immunoblotting assay presenting the phosphorylated STAT5B (P-STAT5B) and SOCS5 in nonviruliferous and viruliferous SBPHs. (E) Analysis of P-STAT5B and SOCS5 in SBPHs during RSV infection. Insect samples at 1, 2 and 4 days after injection of RSV crude extracts were subjected to immunoblotting using the indicated antibodies. Injection of nonviruliferous crude extracts as a mock inoculation. ACTB was used as a protein loading control. (F) Relative RNA level of RSV NP in SBPHs at 1, 2 and 4 days after injection of RSV crude extracts as determined by qRT-PCR. (G-I) Relative transcript levels of *SOCS5* and *STAT5B* in nonviruliferous SBPHs at 1, 2 and 4 days after injection of RSV were measured by qRT-PCR. The mock inoculation group was injected with nonviruliferous crude extracts. Experiments were performed three independent replicates. N.S., no significant. **, $p < 0.01$ by the student t-test.

SBPHs. Similar to previous observations, the protein levels of P-STAT5B and SOCS5 significantly increased and BCL2 accumulation was reduced remarkably (Fig 6A, lanes 1–3 vs. 4–6). In addition, knockdown of *STAT5B* or *SOCS5* also significantly reduced RBSDV acquisition

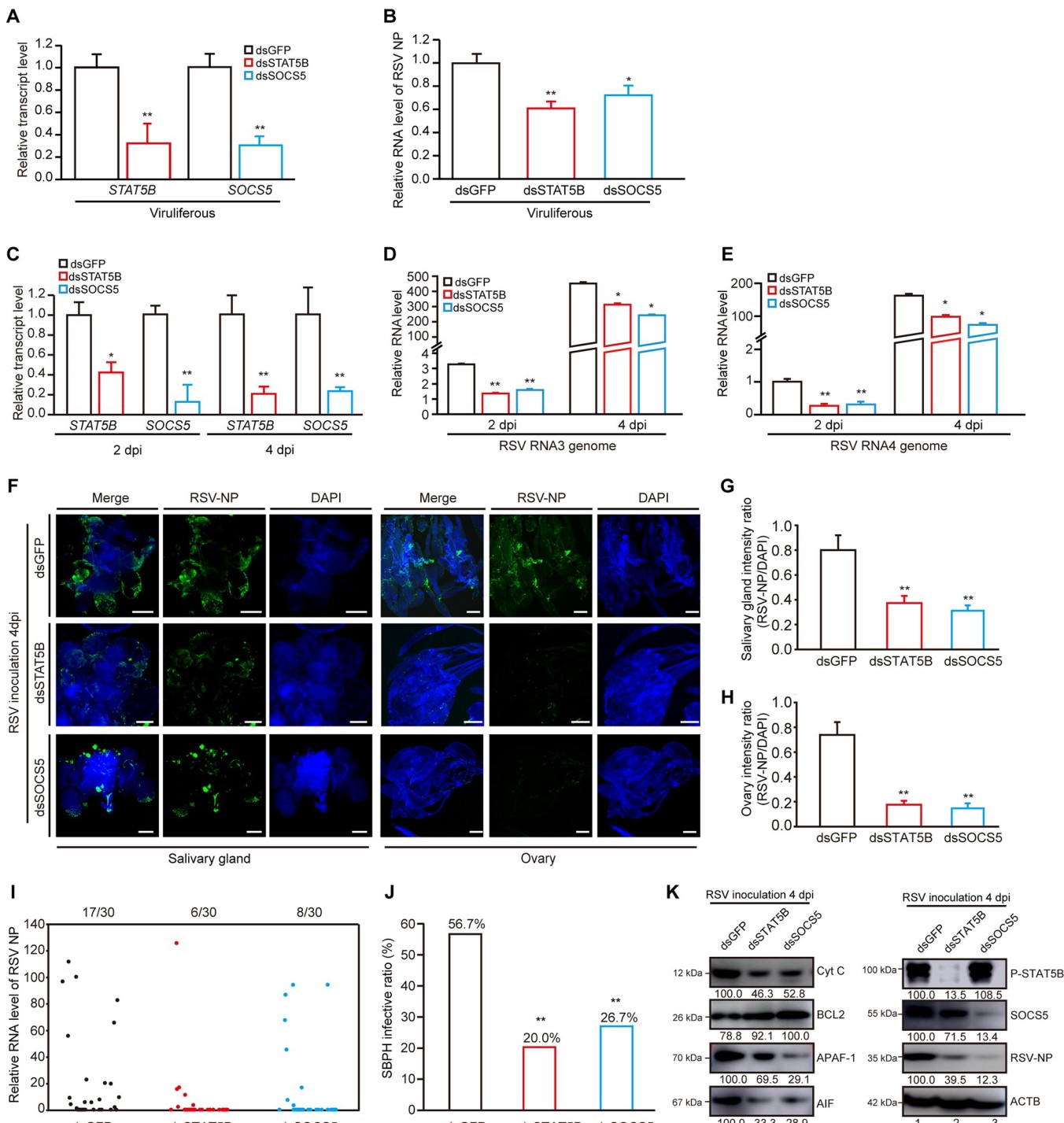

**Fig 5. JAK-STAT pathway regulates apoptosis to promote viral infection.** (A) Relative transcript levels of *STAT5B* and *SOCS5* in viruliferous SBPHs after silencing *STAT5B* and *SOCS5* for 48 h as determined by qRT-PCR. Injection of ds*GFP* into viruliferous SBPHs was acted as the control group. (B) Quantitative RT-PCR analysis of RSV NP in viruliferous SBPHs after silencing *GFP*, *STAT5B* and *SOCS5*. (C) Quantitative RT-PCR analysis of *STAT5B* and *SOCS5* in ds*GFP*-, ds*STAT5B* or ds*SOCS5*-treated SBPHs at 2 and 4 days after injection of RSV crude extracts. (D-E) Quantitative RT-PCR analysis of viral genome RNA3 (D) and RNA4 (E) in ds*GFP*-, ds*STAT5B* or ds*SOCS5*-treated SBPHs at 2 and 4 days after injection of RSV crude extracts. (F) Immunolabeling detection of RSV NP (green) in salivary glands and ovaries of ds*GFP*-, ds*STAT5B* or ds*SOCS5*-treated SBPHs at 4 days after injecting RSV crude extracts. Nuclei were stained with DAPI (blue). Bar, 100 μm. (G-H) Fluorescence density of RSV NP signal in salivary glands (G) and ovaries (H). Relative intensity was quantified by ImageJ software. The experiment was replicated three times. (I) Quantitative RT-PCR analysis of RSV NP in each SBPH. The number of RSV-infected SBPHs relative to total SBPHs is shown at the top of column. Each dot represents a SBPH sample. (J) Differences in SBPH infective ratios were compared at 4

days after injecting RSV crude extracts. *, p < 0.05 and **, p < 0.01 by the student t-test. (K) Immunoblotting analysis of apoptotic response in ds*GFP*-, ds*STAT5B* or ds*SOCS5*-treated SBPHs at 4 days after injection of RSV crude extracts. ACTB was used as a protein loading control.

by SBPHs (Fig 6B and 6C). In general, our results indicated that JAK-STAT pathway may also promote other plant viral infection in SBPHs.

## Discussion

Arboviruses need to manipulate innate immune system to establish persistent viral infection in insect vectors. In this study, we demonstrate that two plant viruses (RSV and RBSDV) activate JAK-STAT signaling pathway in an insect vector. The SOCS5, an E3 ubiquitin ligase regulated by STAT5B, directly interacts with anti-apoptotic BCL2 and accelerates the degradation of BCL2 through the 26S proteasome pathway. This process allows the release of cytochrome *c* from mitochondria into cytosol and induces subsequent apoptosis, thereby enhancing viral replication (Fig 7).

The JAK-STAT signaling pathway has been extensively reported to be involved in antiviral immunity in mammalian and invertebrate systems. For example, interferon-induced JAK-STAT signaling is activated in response to Hepatitis C virus, Chikungunya virus or Sindbis virus, and viral proteins modulate STAT1 protein expression for viral infection in mammalian cells [55–57]. In mosquito vector, dengue virus (DENV), Zika virus, West Nile virus or yellow fever virus can activate JAK-STAT signaling and the pathway inhibition benefits viral proliferation [17,58–60]. In *Drosophila*, JAK-STAT pathway protects against *Drosophila* C virus, cricket paralysis virus or invertebrate iridescent virus 6 and loss-of-function mutant flies associated with increased susceptibility to viral infection [61,62]. On the contrary, a few studies have indicated that the JAK-STAT pathway regulator STAT5 activates two DNA damage pathways to promote human papillomavirus replication and the knockdown of STAT5 abolishes viral genome amplification in mammalian cells [63,64]. Similarly, infection of white spot syndrome virus (WSSV) increases the phosphorylated STAT, while the disruption of STAT activation decreases WSSV infection in shrimp [65,66]. Our study showed that viral infection activated STAT5B phosphorylation, accelerated BCL2 consumption and triggered subsequent apoptosis response. This process represents a novel mechanism that virus activates innate immune system to promote persistent infection.

It is intriguing to figure out how viral infection activates STAT5B phosphorylation in SBPH vectors. Although the unpaired ligand UPD and transmembrane receptor Domeless (Dome) are reported in mammal and other insects, the UPD-like or Dome-like orthologs have not been identified in SBPHs, suggesting that additional cytokine receptors may exist to active STAT5B phosphorylation. A recent study in *Bombyx mori* indicates that C-type lectin 5, instead of Dome, interacts with hopscotch and may be a new pattern recognition receptor to activate JAK-STAT signaling [67]. Whether there is a similar mechanism to active STAT5B phosphorylation in response to viral infection requires further investigation. In addition, many immune pathways, including siRNA, JNK, AMPK, and Toll pathway, have been reported to be involved in viral infection in SBPHs [34,41,45,68]. An interesting study will concentrate on the crosstalk between pathways to promote persistent viral infection.

It has been considered that SOCS family proteins are mainly involved in the negative feedback regulation of the JAK-STAT pathway [69,70]. Our results also demonstrated that the protein level of phosphorylated STAT5B increased after silencing of SOCS5 (Fig 5K). SOCS family proteins have been reported to be hijacked by multiple viruses to inhibit the expression of interferon and to facilitate viral infection in mammal hosts [26]. Whether RSV or RBSDV can hijack SOCS5 to escape antiviral immunity in insect vectors remains to be further

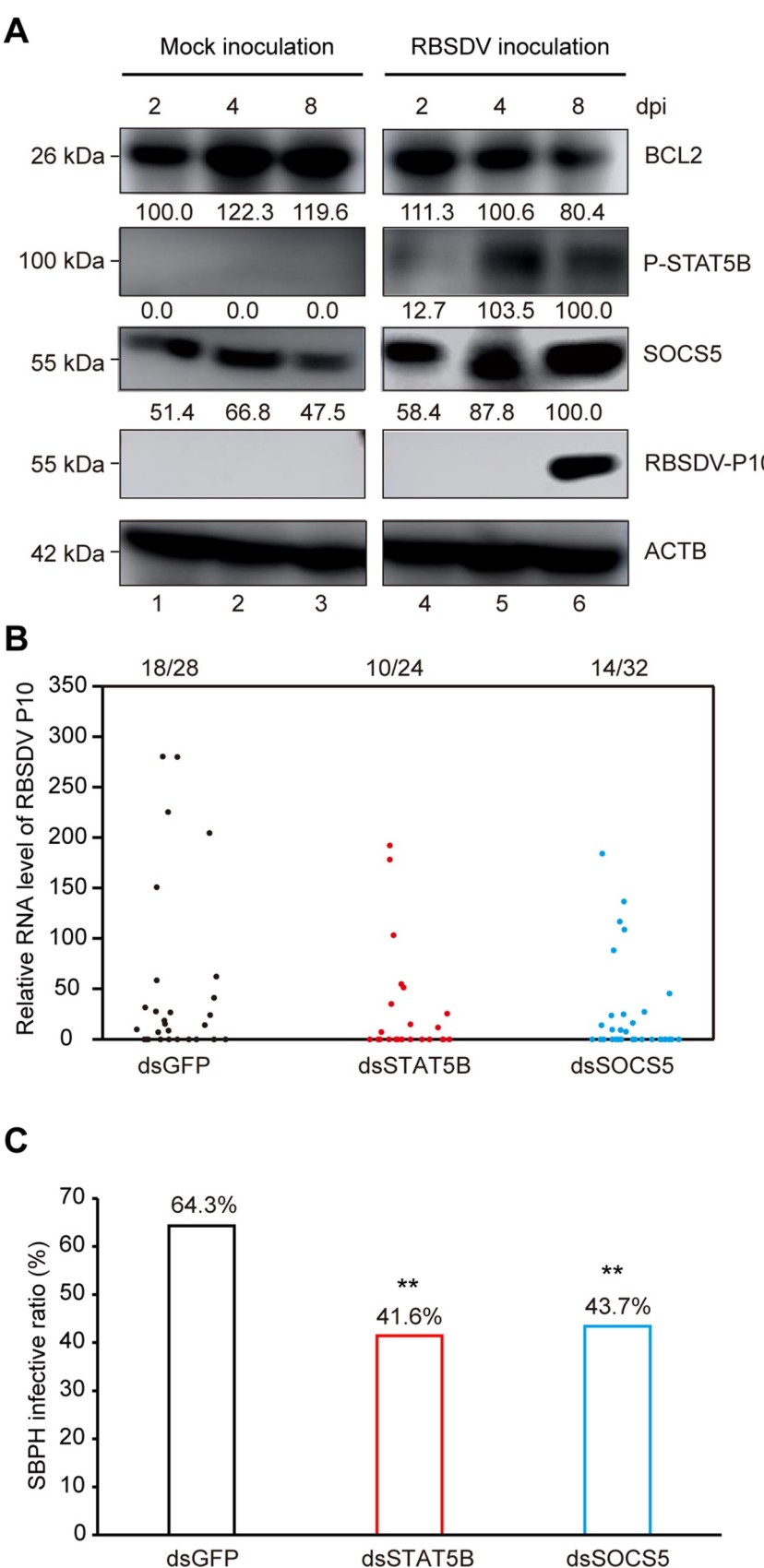

**Fig 6. JAK-STAT pathway promotes other plant viral infection in insect vectors.** (A) Apoptotic response and JAK-STAT pathway were trigged in SBPHs during RBSDV infection. ACTB acted as a protein loading control. (B) Relative RNA level of RBSDV P10 in each SBPH as determined by qRT-PCR. The number of RBSDV-infected SBPHs relative to total SBPHs is shown at the top of column. Each dot represents a SBPH sample. (C) Differences in SBPH infective ratios were compared at 8 days after injecting RBSDV crude extracts. **, $p < 0.01$ by the student t-test. Experiments were performed three independent replicates.

explored. Moreover, we have found that SOCS5 can directly interact with BCL2 and promote the degradation of BCL2 through the 26S proteasome pathway. This result provides a novel function for SOCS family in the manipulation of apoptosis.

Recent studies have shown that the crosstalk between apoptosis and autophagy plays an important role in maintaining persistent viral infection. In *A. aegypti*, apoptosis-related caspase regulated autophagy and affects DENV infection [11]. Moreover, a phosphatidylethanol-amine-binding protein balances autophagy and apoptosis to control viral load in whitefly [37]. The infection with southern rice black-streaked dwarf virus triggers BCL2 interacting protein-mediated mitophagy and attenuates mitochondria-dependent apoptosis in white-backed planthopper [71]. A recent study reports that the early phase of RBSDV infection induces autophagy to inhibit viral accumulation in SBPHs [45]. Our results demonstrated that the infection with RBSDV triggered SOCS5-mediated apoptosis to benefit viral replication. The future study can uncover the host factors balancing that apoptosis and autophagy in virus-vector coexistence.

In summary, we have found that two plant viruses activate the JAK-STAT pathway to promote persistent viral infection in insect vector, and uncover a new mechanism for the

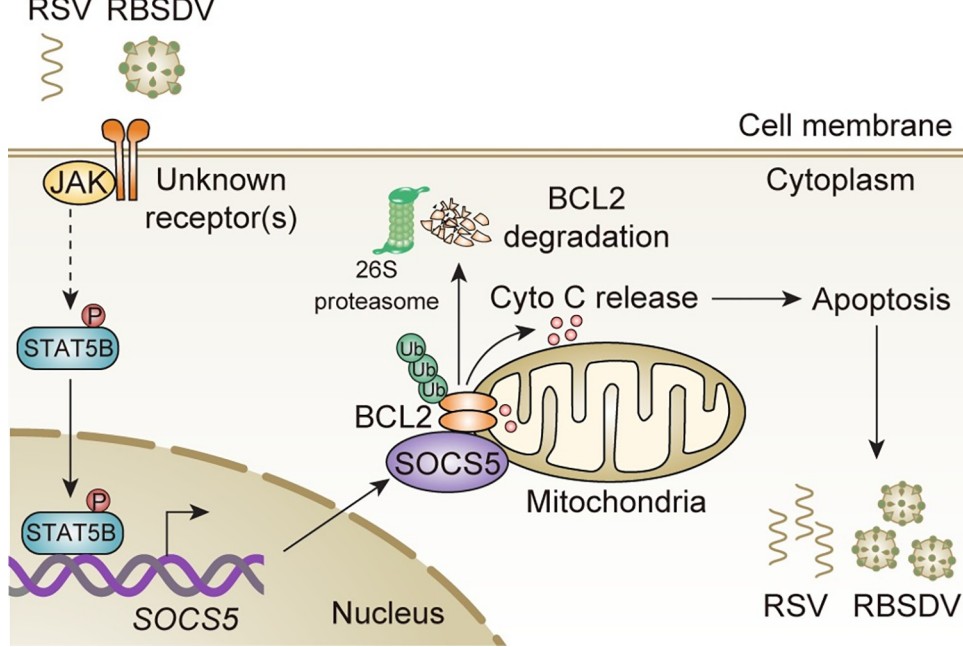

**Fig 7. The model of JAK-STAT pathway regulates apoptosis during viral infection.** Two plant viruses (RSV and RBSDV) activate JAK-STAT signaling pathway in their SBPH vector. The SOCS5, an E3 ubiquitin ligase regulated by STAT5B, directly interacts with anti-apoptotic BCL2 and accelerates the degradation of BCL2 through the 26S proteasome pathway. This process allows the release of cytochrome *c* from mitochondria into cytosol and induces subsequent apoptosis, resulting in the enhancement of viral replication. How the STAT5B is phosphorylated by the JAK in SBPHs remains to be explored.

regulation of apoptosis by innate immune system. Moreover, this mechanism can also be employed to other persistent arboviruses in their insect vectors and the inhibition of JAK-STAT pathway may serve as an effective antiviral strategy to block the transmission of persistent arboviruses.

## Materials and methods

### Source of virus, plant and insect vector

The viruliferous and nonviruliferous small brown planthopper were reared separately on rice seedlings in a growth chamber at 26±1°C with 80% relative humidity and 14 h light/ 10 h dark photoperiod. The infection ratio of viruliferous population was approximately 90% and monitored every 3–5 generations by RT-PCR according to previous description [44]. The RSV-infected and RBSDV-infected rice plants were obtained from Jiangsu Province, China, and stored in the laboratory.

### Double-stranded RNA synthesis and injection

The dsRNAs targeting approximately 500 bp regions of *Casp1a*, *Casp1c*, *Casp8*, *CaspNc*, *SOCS5*, *STAT5B* and *GFP* were synthesized by adopting the T7 High Yield Transcription Kit (Vazyme, China) in line with the manufacturer's protocols. The corresponding PCR primers with T7 promoter sequences for these genes are available in S1 Table. The quality of synthesized dsRNA was determined using agarose gel electrophoresis. Each third-instar nymphs were injected with 25 nl of dsRNA into hemolymph in the insect ventral thorax via a glass needle based on a TransferMan 4r micromanipulator (Eppendorf, Hamburg, Germany).

### Injection of virus crude extracts

Approximately 10 adult viruliferous insects were homogenized with a disposable polypropylene pestle in 100 μL of 10 mM PBS (pH 7.4) in a 1.5 mL EP tube. After centrifugation at 8,000 rpm for 5 min at 4°C, the supernatant was retained. Centrifugation was repeated three times, and the supernatant from the last centrifugation was applied as the viral crude extracts. A total of 25 nL of virus crude extracts was injected into the hemolymph of nonviruliferous third-instar nymphs using a TransferMan 4r micromanipulator. Injection of 25 nl of crude extracts from nonviruliferous insects was applied as a mock inoculation.

### RNA extraction and quantitative real-time PCR

Total RNA was extracted from SBPH insects using the TRIzol Reagent (Invitrogen, Carlsbad, CA, USA). The RNA quality was evaluated using a NanoDrop 2000 spectrophotometer (Thermo Fisher Scientific, USA). Total RNA was subjected to reverse transcription to cDNA based on HiScript II Q RT SuperMix for qPCR (+gDNA wiper) (Vazyme, China). Quantitative real-time PCR (qRT-PCR) was performed on a LightCycler Real-Time PCR System (Roche, Swiss) with Hieff qPCR SYBR Green Master Mix (Yeasen, China). The expression level of each gene in SBPH was normalized to the expression of two housekeeping genes (*ACTIN* and *GAPDH*) and was calculated with the $2^{-\Delta\Delta Ct}$ method. Three biological replicates were conducted for each experiment and two technical replicates were set up for each biological replicate.

### Immunofluorescence labeling

Immunofluorescence labeling was performed as previously described [43] with slight modification. SBPH samples were dissected in 1×PBS (pH 7.4) and fixed in 4% paraformaldehyde

solution for 1 h. After washing three times with 1×PBS (pH 7.4), the samples were treated with the 2% Triton X-100 solution for 0.5 h and subsequently incubated in the 1:200 (v/v) diluted FITC-conjugated anti-NP specific antibody for 2 h at room temperature. Then, the samples were washed three times in PBS and mounted in Fluoroshield mounting medium with DAPI (Abcam, UK). The mounted samples were viewed using a Leica TCS SP8 confocal microscope (Leica Microsystems, Solms, Germany). Relative intensity of RSV-NP and DAPI was analyzed using ImageJ software.

### Terminal uridine nick-end labeling (TUNEL) assay

The DeadEnd Fluorometric TUNEL System (G3250, Promega, USA) was used for TUNEL staining. 15 insect midguts and 15 salivary glands were dissected in 10 mM PBS (pH 7.4) and fixed in 4% paraformaldehyde for 0.5 h. After three rinses in PBS, samples were incubated with rTdT incubation buffer at 37˚C for 60 min in dark. The tissues were mounted in Fluoroshield mounting medium with DAPI (Abcam, UK) and detected under a Leica TCS SP8 confocal microscope (Leica Microsystems, Solms, Germany).

### Transmission electron microscopy

In this study, tissue samples obtained from viruliferous or nonviruliferous were dissected and fixed with 2.5% glutaraldehyde solution for 4 h and fixed with 1% OsO4 for another 2 h. After dehydration using ethanol with increasing concentration, intestines were treated with acetone for 1 h, followed by embedding in LX-112. The midguts were cut using Leica EM UC7 Ultramicrotome and stained with uranyl acetate and alkaline lead citrate. Samples were observed under a Hitachi HT-7800 TEM (Hitachi, Tokyo, Japan).

### Protein expression, purification and MBP pull-down

Recombinant proteins of MBP, MBP-BCL2, full-length and fragments of His-SOCS5 were transformed into the *E. coli* stain BL21. To express recombinant proteins, 0.5-L culture was grown at 37˚C until an OD600 of 0.8 in a shaker incubator. Proteins were induced with 0.1 mM isopropyl-D-thiogalactopyranoside (IPTG) for 12 h at 16˚C. MBP- or His-fused recombinant proteins were purified using Amylose Resin beads (E8021, NEB, USA) or nickel-nitrilotriacetic acid resin (Ni-NTA, Qiagen, Germany) as instructed by the manufacturer, respectively.

For MBP pull-down, purified MBP or MBP-BCL2 proteins were initially incubated with Amylose Resin beads (E8021, NEB, USA) for 2 h at 4˚C. After being washed three times with lysis buffer, two beads were individually inocubated with wild-type and mutants of His-SOCS5 for another 4 h at 4˚C. Finally, these eluates were collected for Western blotting analysis.

### Western blotting assay

Protein samples were added with a loading buffer containing sodium dodecyl sulfate (SDS) and boiled at 95˚C for 10 min. Then samples were separated in 10% (w/v) SDS-PAGE gels and transferred onto polyvinylidene fluoride (PVDF) membranes. The membranes were probed with a specific primary antibody diluted at 1:5,000 (v/v), followed by incubation with a horseradish peroxidase (HRP)-conjugated goat anti-rabbit/mouse antibody diluted at 1:10,000 (v/v). The blots were visualized using the Luminescent Image Analyzer AI680 (GE, Sweden). Relative band intensity was caculated by ImageJ sofeware.

Mouse polyclonal antibody against RSV NP was produced in our laboratory. Phospho-STAT5 anti-rabbit IgG was obtained from Cell Signaling Technology (9351, Danvers, MA,

USA). BCL2 anti-rabbit IgG (ET1702-53), Cytochrome c anti-rabbit IgG (R1510-41), APAF-1 anti-rabbit IgG (ET1607-12), AIF anti-rabbit IgG (ET1603-4), SOCS5 anti-rabbit IgG (HA500307) and ACTB (beta-actin) anti-mouse IgG (EM21002) were obtained from Huabio (Hangzhou, China). MBP-tag anti-mouse IgG (MA5-14122) and His-tag anti-mouse IgG (MA1-21315) were acquired from Invitrogen (Carlsbad, CA, USA). The detailed information of these commercial antibodies was displayed in S2 Table.

## Yeast two-hybrid interaction

For yeast two-hybrid (Y2H) screen, the Gal4-DNA binding domain (BD)-fused BCL2 in pGBKT7 was employed to transform the yeast strain AH109 (BD Clontech). The transformed AH109 yeast cells were mated with a Gal4-DNA activating domain (AD)-fused small brown planthopper cDNA library in Y187 yeast cells. The putative interaction-positive yeast cells were selected at the high-stringency selective synthetic dextrose (SD) medium lacking trypto-phan, leucine, histidine, and adenine (SD/-T/-L/-H/-A) plates. The selected cDNA fragments were extracted and sequenced.

For one-to-one Y2H assay, the pGBKT7-BCL2 (BD-BCL2) and isolate prey vectors were co-transformed into AH109 yeast cells and selected on the SD medium lacking tryptophan and leucine (SD/-T/-L) for 4 days at 30°C. Subsequently, the yeast cells were transferred to the SD/-T/-L/-H/-A plate to analyze the protein-protein interaction.

## Treatment with inhibitor and inducer

The apoptosis inhibitor Z-DEVD-FMK (MB2588, Meilunbio, China) and inducer PAC-1 (MB4074, Meilunbio, China) were used to examine the effect of apoptotic pathways on RSV infection. 300 nonviruliferous third-instar nymphs were injected with RSV crude extracts and divided into three groups to be treated with a diet of 10 μM Z-DEVD-FMK with 20% sucrose, 10 μM PAC-1 with 20% sucrose, and 20% sucrose containing 0.1% DMSO, respectively, using double-parafilm feeding for 48 h. Then the three groups of insects were transferred to feed on healthy rice seedlings for another 48 h and collected for Western blotting analysis. The treatment of proteasome inhibitor MG132 on viruliferous SBPHs was performed using the same methods as previously described.

## Yeast one-hybrid interaction

The recombinant pABAi plasmids were generated by inserting three promoter sequences of *SOCS5* gene, respectively. The plasmids were then transformed into the yeast strain Y1HGold. After that, the minimum inhibitory concentrations of aureobasidin A (AbA) for normal growth of the bait strains were selected. Prey plasmids were generated into the pGADT7 vec-tor, which were co-transformed into the bait strains yeast cells. The yeast cells were grown on a SD medium lacking leucine (SD/-L) with AbA for 4 days at 30°C.

## Electrophoretic mobility shift assay

Recombinant proteins of His-STAT5B were transformed into the *E. coli* stain BL21. Purified His-STAT5B proteins were initially incubated with Ni-NTA for 2 h at 4°C before detection. 1 μg template DNA was marked with 4 μL DIG-High Prime and incubated for all night at 37°C as instructed by the manufacturer (11585614910, Roche, Switzerland). The DNA frag-ments and purified proteins were incubated at 37°C for 20 min. Later, the electrophoresis at constant pressure of 120 V was conducted on the agarose gel for 40 min, and then transferred onto nylon membrane (Amersham Hybond N+, GE Healthcare, China). The membranes

were labeled with RNA using UV crosslinker and probed with anti-digoxigenin-AP antibody diluted at 1:10,000 (v/v), and the blots were visualized by adopting the Luminescent Image Analyzer AI680 (GE, Sweden).

## Mitochondrial isolation

Extracted and purified mitochondria of insect tissue were performed according to kit instructions (C3606 Beyotime, China). In brief, 100 mg SBPH samples were added with 1 mL Reagent A, homogenized on ice, and centrifuged at 600 g for 5 min at 4˚C to absorb the supernatant. The sample was centrifuged at 11000 g for 10 min at 4˚C and added with 200 μL Reagent B, then mitochondria were isolated. The collected supernatant was centrifuged at 12000 g for 10 min at 4˚C. Besides, the supernatant was the cytoplasmic protein with the removal of mitochondria. Finally, these samples were collected for Western blotting analysis.

## Supporting information

**S1 Fig. Structure feature and phylogenetic analysis of BCL2 (A), Cyt C (B), Casp1a (C), CaspNC (D), Casp1C (E) and Casp8 (F).** Phylogenetic tree analysis with the maximum likelihood method was based on amino acid sequences of *L. striatellus* and other insects.
(TIF)

**S2 Fig. Western blotting analysis of APAF-1, AIF and Cyt C in nonviruliferous SBPHs that fed with PAC-1 or DMSO.** ACTB acted as a protein loading control.
(TIF)

**S3 Fig. Structure feature and phylogenetic analysis of SOCS5.** (A) Schematic diagrams showing SOCS5. (B) Phylogenetic tree analysis with the maximum likelihood method was based on amino acid sequences of SBPH and other insects.
(TIF)

**S4 Fig. Western blotting and qRT-PCR analysis of JAK-STAT pathway genes in SBPHs.** (A) Western blotting analysis of SOCS5 in the ds*JAK*- or ds*GFP*-treated nonviruliferous SBPHs. ACTB was used as a protein loading control. (B) Relative transcript expression of JAK-STAT pathway genes in nonviruliferous and viruliferous SBPH as detected by qRT-PCR. Six genes (*JAK*, *TYK2*, *STAT5B*, *STAM*, *PTPN2* and *PIAS*) from JAK-STAT pathway were detected. The experiment was replicated three times.
(TIF)

**S5 Fig. Structure feature and phylogenetic analysis of JAK (A), TYK2 (B), STAT5B (C), STAM (D), PIAS (E) and PTPN2 (F).** Phylogenetic tree analysis with the maximum likelihood method was based on amino acid sequences of *L. striatellus* and other insects.
(TIF)

**S6 Fig. Effect of RSV infection on the infective ratio and survival ratio in SBPHs after injecting of ds*GFP*, ds*STAT5B* or ds*SOCS5*.** (A) Differences in SBPH infective ratios were compared at 8 days after feeding on RSV-infected rice plants. **, $p < 0.01$ by the student t-test. (B) Survival of ds*GFP*-, ds*STAT5B*- or ds*SOCS5*-treated SBPH after the injection of RSV crude extracts. Each point represents the mean value of three biological replicates. *$P<0.05$ by the student t-test.
(TIF)

**S1 Table. Primers used in this study.**
(DOCX)

**S2 Table. The detailed information of commercial antibodies used in this study.**
(DOCX)

## Author Contributions

**Conceptualization:** Bo-Xue Li, Chuan-Xi Zhang, Gang Lu.

**Data curation:** Yan Zhang, Ji-Chong Zhuo, Hai-Jian Huang, Jia-Bao Lu, Chuan-Xi Zhang.

**Formal analysis:** Yan Zhang, Qian-Zhuo Mao, Jun-Min Li.

**Funding acquisition:** Bo-Xue Li, Hai-Jian Huang, Jian-Ping Chen, Gang Lu.

**Investigation:** Yan Zhang, Qian-Zhuo Mao, Jia-Bao Lu.

**Methodology:** Qian-Zhuo Mao, Ji-Chong Zhuo, Hai-Jian Huang, Jia-Bao Lu, Jun-Min Li.

**Project administration:** Bo-Xue Li, Chuan-Xi Zhang, Jian-Ping Chen.

**Resources:** Ji-Chong Zhuo, Hai-Jian Huang, Jia-Bao Lu, Jun-Min Li, Jian-Ping Chen.

**Supervision:** Bo-Xue Li, Chuan-Xi Zhang, Jun-Min Li, Jian-Ping Chen, Gang Lu.

**Validation:** Jia-Bao Lu, Chuan-Xi Zhang, Jian-Ping Chen, Gang Lu.

**Visualization:** Bo-Xue Li, Ji-Chong Zhuo, Chuan-Xi Zhang.

**Writing – original draft:** Yan Zhang, Gang Lu.

**Writing – review & editing:** Bo-Xue Li, Gang Lu.

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
