## [Decision Letter · Decision Letter 0]

20 Feb 2023

Dear Dr Gang Lu,

Thank you very much for submitting your manuscript "The JAK-STAT pathway promotes persistent viral infection by activating apoptosis in insect vectors" for consideration at PLOS Pathogens. As with all papers reviewed by the journal, your manuscript was reviewed by members of the editorial board and by several independent reviewers. The reviewers appreciated the attention to an important topic. Based on the reviews, we are likely to accept this manuscript for publication, providing that you modify the manuscript according to the review recommendations.

Sincerely,

Xiaorong Tao, Ph.D.

Guest Editor

PLOS Pathogens

Bart Thomma

Section Editor

PLOS Pathogens

Kasturi Haldar

Editor-in-Chief

PLOS Pathogens

orcid.org/0000-0001-5065-158X

Michael Malim

Editor-in-Chief

PLOS Pathogens

orcid.org/0000-0002-7699-2064

Reviewer Comments (if any, and for reference):

Reviewer's Responses to Questions

**Part I - Summary**

Reviewer #1: In this study the authors described the role of JAK-STAT pathway in the persistent viral infection in small brown planthopper. They showed RSV and RBSDV infection activated the JAK-STAT pathway and promoted the accumulation of SOCS5, and accelerated the BCL2 degradation through the 26S 30 proteasome pathway, causing the activation of apoptosis facilitated persistent viral infection in their vector. This is an interesting story about virus-vector interaction and is well-written. I would recommend acceptance after responding to the listed concerns.

Reviewer #2: This is a resubmitted manuscript. The authors have address all of my previous comments.

Reviewer #3: In this manuscript authors described the role of JAK-STAT pathway in promoting persistent viral infection by activating apoptosis in insect vectors small brown planthopper. They showed that both RSV and RBSDV, two important rice viruses transmitted by small brown planthopper, activated the JAK-STAT pathway and promoted the accumulation of Suppressor of cytokine signaling 5 (SOCS5), an E3 ubiquitin ligase regulated by the JAK-STAT transcription factor STAT5B. The manuscript is significantly improved after reversion.

**Part II – Major Issues: Key Experiments Required for Acceptance**

Reviewer #1: 1, The evidences of SOCS5 interacts and degrade BCL2 are relatively less, and there is only on data (Fig 3E) showed how BCL2 was degraded. Is it possible to show BCL2 was ubiquitized by SOCS5 and the ubiquitination is enhanced by virus infection.

2, In Fig 4E and 4G, in 1 dpi, the transcript level of SOCS5 was significantly increased, but the protein level was similar. And in Fig 4F, the PCR showed the RSV level was accumulated at 1, 2, and 4 dpi, but in Fig 4E, RSV-NP was only detected in 4 dpi.

3, Some immunoblot of phos-STAT5B was low quality, like in Fig 4D and 4E, one has a specific one band, and the other has two bands.

Reviewer #2: (No Response)

Reviewer #3: No

**Part III – Minor Issues: Editorial and Data Presentation Modifications**

Reviewer #1: 1, There is only Fig 6 showed RBSDV may has similar mechanism, the data is relatively weak and I think the authors has over-stated the conclusion.

2, Line 235, there are two “STAT5B”, please delete one.

Reviewer #2: (No Response)

Reviewer #3: (No Response)

PLOS authors have the option to publish the peer review history of their article (what does this mean?). If published, this will include your full peer review and any attached files.

Reviewer #1: **Yes: **Tong Zhang

Reviewer #2: No

Reviewer #3: **Yes: **Yi Xu

Figure Files:

Data Requirements:

Reproducibility:

References:

---

## [Editor Report · Decision Letter 1]

4 Mar 2023

Dear Dr.Lu,

We are pleased to inform you that your manuscript 'The JAK-STAT pathway promotes persistent viral infection by activating apoptosis in insect vectors' has been provisionally accepted for publication in PLOS Pathogens.

Best regards,

Xiaorong Tao, Ph.D.

Guest Editor

PLOS Pathogens

Bart Thomma

Section Editor

PLOS Pathogens

Kasturi Haldar

Editor-in-Chief

PLOS Pathogens

orcid.org/0000-0001-5065-158X

Michael Malim

Editor-in-Chief

PLOS Pathogens

orcid.org/0000-0002-7699-2064
---

## [Editor Report · Acceptance letter]

13 Mar 2023

Dear Dr. Lu,

We are delighted to inform you that your manuscript, "The JAK-STAT pathway promotes persistent viral infection by activating apoptosis in insect vectors," has been formally accepted for publication in PLOS Pathogens.

Best regards,

Kasturi Haldar

Editor-in-Chief

PLOS Pathogens

orcid.org/0000-0001-5065-158X

Michael Malim

Editor-in-Chief

PLOS Pathogens

orcid.org/0000-0002-7699-2064